# Mobile Dressing Trolleys Improve Satisfaction and Logistics on Pediatric Surgery Wards

**DOI:** 10.3390/children10071089

**Published:** 2023-06-21

**Authors:** Hannes Franck, Astrid Dempfle, Katja Reischig, Jonas Baastrup, Andreas Meinzer, Meike Kossakowski, Thomas Franz Krebs, Robert Bergholz

**Affiliations:** 1Department of General, Visceral, Thoracic, Transplant and Pediatric Surgery, UKSH University Hospital of Schleswig-Holstein Kiel Campus, Arnold-Heller-Strasse 3, 24105 Kiel, Germany; 2Institute of Medical Informatics and Statistics, UKSH University Hospital of Schleswig-Holstein Kiel Campus, Arnold-Heller-Strasse 3, 24105 Kiel, Germany; 3Department of Pediatric Surgery, Children’s Hospital of Eastern Switzerland, Claudiusstrasse 6, 9006 St. Gallen, Switzerland

**Keywords:** surgery, pediatric surgery, dressing changes, logistics, patient’s satisfaction, recovery after surgery

## Abstract

Background: Evidence-based data on the effect of dressing trolleys on children’s postoperative recovery are not available. The aim of this study was to evaluate a specific pediatric surgical dressing trolley on patient and caregiver satisfaction, as well as temporal and logistical aspects of the dressing change procedures. Methods: In a prospective observational non-randomized study, a total of 100 dressing changes were observed before (group 1) and after (group 2) the introduction of a pediatric surgical dressing trolley and the satisfaction, time and logistical factors were recorded on site. Results: The median preparation time, the duration of the dressing change and the total time decreased significantly from group 1 to group 2 by 1:11 min (*p* < 0.001); 1:56 min (*p* = 0.05) and 5:09 min (*p* = 0.001), respectively. The patient’s room was left significantly less often in group 2 to retrieve missing bandages. The median satisfaction of the medical staff increased by 12% in group 2 (*p* < 0.001). The satisfaction of the parents increased by 2.5% in group 2 (*p* = 0.042), and that of the nursing staff increased by 9.25% in group 2 (*p* = 0.015). Conclusions: Our results demonstrate the positive effects of a dressing trolley for pediatric surgical dressing changes by minimizing postoperative handling and manipulation of the child. It improves time and logistical factors as well as the satisfaction of those involved, which may lead to a faster recovery.

## 1. Introduction

Undergoing surgical intervention is an arduous event that impacts both the children and their caretakers. The subsequent phase of postoperative recovery, which involves the management of wounds and the changing of dressings in delicate and potentially painful areas, plays a pivotal role in the overall healing process [1]. While efforts have been made to mitigate perioperative stress in children, the specific impact of dressing changes on pediatric patients remains unexplored [2].

The literature has touched upon the utilization of dressing trolleys or carts for wound care, with discussions on hygiene aspects dating back to as early as 1958 [3,4]. However, data pertaining to the utilization of dressing trolleys in the context of pediatric surgery is currently absent. Consequently, the objective of this study was to assess the subjective satisfaction levels of pediatric patients and their caregivers both prior to and following the implementation of a specially designed pediatric surgical dressing trolley. Additionally, our study aims to determine whether the introduction of such a trolley would enhance the logistical aspects of these procedures.

## 2. Materials and Methods

The prospective observational study “Quality assurance of nursing, medical, patient-relevant and logistical aspects in the context of the introduction of a pediatric surgical dressing trolley” was approved by the local ethics committee of the Christian-Albrechts-University Kiel (D530/19) and performed at the University Hospital of Schleswig-Holstein (UKSH), Kiel Campus, Germany from 22 October 2019 to 11 September 2020. Parental informed consent was achieved as well as from the medical personnel participating in the study.

This study involved the inclusion of pediatric surgery inpatients, specifically children below 18 years of age, who required postoperative wound dressing changes. The Exclusion criteria encompassed cases where patient or parental consent was lacking; children staying in the intensive care unit; dressing changes performed in the operating room under general anesthesia; and dressings, such as skin glues, that did not necessitate regular dressing changes.

### 2.1. Observational Periods

After physical migration of the pediatric wards from the former children’s hospital into the newly erected University Hospital building in summer 2019, surgical dressing trolleys were introduced for the pediatric surgical wound care of children who were going to be distributed across three newly designed pediatric wards.

As the focus was set on collecting data under everyday clinical conditions, a prospective observational study over two time periods was conceptualized: before (group 1) and after (group 2) the introduction of the dressing trolley.

#### 2.1.1. First Time Period: No Dressing Trolley, Group 1

During the first period, dressing changes were performed at the patients bedside in the child’s room or a separate examination room. All materials had to be collected by the nursing staff in advance from separate storage rooms or the examination room, where the preparation and assembly of the dressing materials usually took place. These storage rooms were in close proximity to all three pediatric stations.

#### 2.1.2. Second Time Period: Application of the Dressing Trolley, Group 2

The introduction of the mobile dressing trolley, which incorporated the main materials needed for different dressing changes marked the begin of the second time period. Therefore, the two groups were acquired during distinct time periods.

### 2.2. The Dressing Trolley and Logistics

The dressing trolley was taken from Schmitz (SCHMITZ u. Soehne GmbH & Co. KG, Wickede, Germany, Figure 1). The trolley had seven drawer compartments of different sizes. The drawers were numbered from one to seven in a clockwise direction and labeled accordingly. The contents of the compartments were recorded on a loading list. This was attached to the dressing trolley in the form of laminated DIN A5 cards. In order to ensure good manageability and clarity of the individual compartments of the dressing trolley, we decided not to store special dressing materials that were rarely used in the trolley. The upper compartments one and seven were equipped with the material that was often used for dressing changes. The larger lower compartments contained special dressing material, which were rarely used, and large products, such as disposable protective gowns and pads (Table 1). Therefore, in the case of complex or special dressing changes (e.g., vacuum-assisted closure (VAC) dressing), the additionally required material was retrieved from the storage rooms, the operating theater or the children’s emergency room by the specialist nursing staff or medical staff, and the time was recorded.

**Figure 1 children-10-01089-f001:**
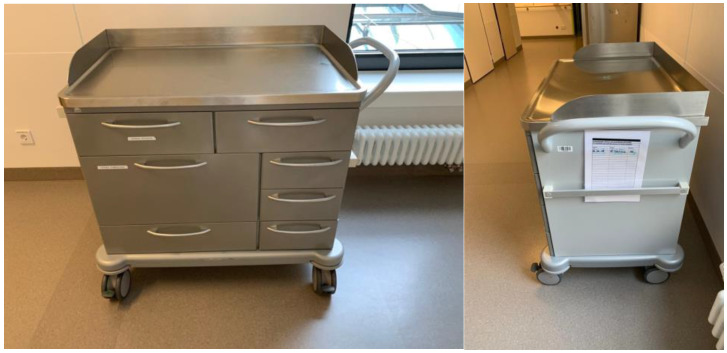
Photographs of the trolley used in this study. The trolley is made of metal, which improves its disinfection. Also of note is the disinfection calendar on the short side of the trolley. The contents are stored in drawers and sealed, see Figure 2.

For the second period, there was no preparation for the dressing changes in comparison to the first period. Instead, the dressing trolley was retrieved from the examination room by the nursing staff and brought to the patient’s room. The dressing trolley was provided and cared for by the first author. If children were housed in isolation rooms for infectious reasons, the trolley was left in the hallway in front of the child’s room.

For the dressing changes, the child, the accompanying parent, a specialist nurse, surgical staff and the first author were present. In addition and on occasion students in their final and practical year, nurse trainees or a second parent were also participating or observing the dressing changes.

### 2.3. Outcome Parameters and Data Acquisition

The primary outcome parameter was the subjective satisfaction with the dressing change (visual analogue scale (VAS) 0–100 mm) of the accompanying parents with or without application of the dressing trolley. The secondary outcome parameters were collected as the following data: 

The gender of the patient, the gender of the parents present, the type and date of the surgery, and the sedation of the child for the dressing change and the occurrence of multiresistant bacteria.

The anticipated mood of the child undergoing the dressing change taken after the procedure by a visual analog scale of the parents or, if the parents were not present, the specialist nurses. 

The general satisfaction with the dressing change (visual analogue scale, VAS, 0–100 mm) of the children (if older than 12 years), nursing staff, surgeons and the observer. The left and right end the VAS was labeled with the words “unsatisfied” and “satisfied”, respectively (Figure 3).

The time (seconds) for the preparation of the dressing change. Only the time spent looking for and depositing the dressing materials was recorded. The time taken to cover the distance between the store rooms and the patient’s room was not recorded because the dressing trolley was also stationed in the store rooms and had to cover the same distance when moved into the patients rooms.

The duration (seconds) of the dressing change.

The occurrences (n) the personnel had to leave the patients room to retrieve additional material that was missing for the dressing change.

The time (seconds) during which the personnel was outside the room to retrieve missing materials.

The occurrences (n) of any interruption of the dressing change.

During time period two, a questionnaire concerning the general load, cleanliness, manageability and overall evaluation of the dressing trolley was answered by the medical staff performing the dressing change. The questions were rated according to the German school grading system 1 (very good) to 6 (bad) (Table 2).

The use of the dressing trolley (yes/no), the location of the dressing change (patient’s room, examination room) and the surgeon performing the dressing change.

The type of dressing applied was classified according to the size of the wounds: Small dressings were a size of up to 5 cm and a maximum of 3 dressings used(e.g., for laparoscopic procedures). Large dressings were a size of 12 cm (e.g., median laparotomy). Special types of dressings comprise those that could not be included in the two abovementioned categories (e.g., vacuum-assisted closure (VAC)).

The category of the surgical procedure (1 = single incision laparoscopic surgery (SILS); 2 = multiport laparoscopy/robotics; 3 = umbilical minilaparotomy; 4 = median/transverse laparotomy; 5 = inguinal hernia, lymph node biopsy/abscess drainage, (sub)cutaneous tumor extirpation/biopsy, burns and anorectal malformation repair; 6 = vacuum-assisted closure (VAC)).

The time for each variable was taken with a smartphone timer app (Multistoppuhr APP for IOS or the APP Multitimer for Android, www.multitimer.net, accessed on 12 August 2019).

### 2.4. Statistical Analysis

The sample size was calculated for the unpaired *t*-test. The expected power was assumed to be 0.8 with an alpha level/significance level of 0.05. With regard to the effect size (Cohen’s d), we expected a clear temporal and logistical advantage after the introduction of the dressing trolley and assumed a medium-to-large effect size. The calculated sample size for d = 0.5 (medium effect) was 64, and for a large effect (d = 0.8), it was 26. Thus, the sample should include at least 26 and ideally 64 dressing changes per observation period. Which resulted in a planned total of minimum 52 to maximum 128 planned dressing changes in both observation periods.

The median and quartiles were used for numeric skewed data. The non-parametric Mann–Whitney U-Test was used to test the logistical aspects and the satisfaction levels. For nominal data, the Chi-square test was used in contingency tables. A *p*-value of *p* < 0.05 was considered significant.

## 3. Results

### 3.1. Outcome Parameters

A total of 100 dressing changes were evaluated (see Table 3 for the baseline, primary and secondary outcome parameters).

### 3.2. Surgeon Performing the Dressing Changes

The surgeons R.B., K.R. and J.B. performed the dressing changes, together with two pediatric surgery residents. There was no significant difference between the two groups of the relation of surgeons performing the dressing change (*p* = 0.22).

### 3.3. Subjective Evaluation of the Dressing Trolley

Of the 49 dressing changes in group 2, the questionnaire was answered in 47 cases (11 were completed by the nursing staff and 36 were completed by the surgeons who performed the procedure). The dressing trolley was given an overall grade of 1.4, see Table 4.

## 4. Discussion

We were able to demonstrate that introducing a mobile dressing trolley into the routine dressing changes on pediatric surgical wards minimizes postoperative handling and thus postoperative trauma to the pediatric patient. It furthermore increases the satisfaction of the caregivers and medical personnel and enhances the overall logistical factors with the procedures.

### 4.1. Satisfaction with the Procedures

In contrast to the other results, the satisfaction of the children did not increase significantly after introduction of the dressing trolley. Determining child satisfaction in this study turned out to be complicated. We determined that children could independently express their satisfaction if they were able to do so based on their age and their cognitive development, but the average age group in our study were toddlers. In group 1 the median age was 1.97 years, and in group 2, it was 2.64 years. Therefore, the satisfaction of the children could only be determined in 9 of 51 children in group 1 and in 8 out of 49 children in group 2. To address this, it would be necessary to include older children in future studies.

Visual analog scales (VAS) were chosen to record patient and caregiver satisfaction after the dressing change had been completed. Data were recorded from children who were able to answer the VAS based on their age and cognitive development, as judged by the surgeons and first author. These were most often children who were older than 6 years [5]. A child’s age of over 5.6 years and an expected IQ of >100 correlated best with the probability that the children had understood the use of the VAS and were able to document their subjective feelings using it [6,7]. 

Recording the satisfaction with the dressing changes consists of measuring a subjective sensation that is, understandably, dependent on many influencing factors. This subjective sensation was objectified by a visual analogue score (VAS). Its advantage lies in its ease of use and the wide range of applications. In addition, the VAS was accepted by the patients as a recording method and could be processed quickly. Furthermore, during the recording of chronic pain in patients with rheumatoid arthritis, good test–retest reliability, good validity and the ability to determine changes in pain or satisfaction could be seen in the VAS [8]. Interestingly, Ari Voutilainen and colleagues research titled “How to ask about patient satisfaction? The visual analogue scale is less vulnerable to confounding factors and ceiling effect than a symmetric Likert scale” reported on 150 patients from the five largest tertiary hospitals in Finland, who were evaluated after their stay with visual analogue scales and Likert scales regarding their satisfaction, with the result that the VAS was not very susceptible to cofounding factors and was useful in practice due to its short processing time [9].

### 4.2. The Effect of Stress on the Surgical Outcome and the Influence of Patient Satisfaction 

Stress is an important factor for patient outcome after surgery. A study by Kiecolt-Glaser et al. demonstrated that a wound after a defined punch biopsy in patients who were psychologically stressed by caring for relatives healed significantly more slowly, and less interleukin 1 beta could be detected in the blood than in the unexposed comparison group [10]. In 2003, Elisabeth Broadbent conducted a study in a clinical setting to examine the influence of stress and anxiety on wound healing. Before the operation, a standardized questionnaire was used to record the stress level and fears about the upcoming operation. In addition, the interleukin-1 and 6, metalloproteinase-9 levels in the wound secretion were recorded. The results demonstrated that in patients with a high level of stress and anxiety, the wounds healed more slowly, and the wound healing-promoting factors in the wound secretions were reduced [11]. In 2006, Lynanne McGuire et al. examined the connection between postoperative pain and the wound healing of a standardized 2 mm punch biopsy in women after elective gastric bypass operations. They came to the conclusion that there is a significant connection between postoperative pain and slower wound healing [12].

The increase in satisfaction in group 2 and the proposed reduction in stress were therefore not only beneficial for the external image and reputation of the department but may also contribute significantly to the recovery of the patients.

### 4.3. Increasing Patient, Parent and Medical Staff Satisfaction 

In order to increase satisfaction and therefore reduce stress, some points have to be discussed. Patients and, in the case of children, their parents too, have expectations of their medical care: What is most important is the restoration of health. This was followed by professional and competent medical and nursing care, which, ideally, should be carried out respectfully. In addition, empathy as well as sufficient and understandable information were desired, and points such as professional competence and, for example, hygiene, were taken for granted. This assumption reaches back to the Herzberg two-factor theory, a model that was originally applied in job satisfaction [13]. It assumes that satisfaction is divided into two factors: hygiene and motivational factors. Hygiene factors are services that are taken for granted. The fulfillment of these services with the medical intervention provided was a prerequisite for the satisfaction of a patient or their relatives. Hygiene factors prevent dissatisfaction yet do not create satisfaction; motivational factors are responsible for this. That is, these were services that were necessary to create “satisfaction”, but when they were absent, they did not generate dissatisfaction. The use of hospital clowns acts as an example of such a motivational factor in children’s wards. They can help the children to make everyday hospital life more pleasant and thus lead to an increase in satisfaction. Due to the fact that they were not available as a “standard” in all hospitals, a lack of clowns would not necessarily lead to dissatisfaction [14].

Dressing changes are suitable motivational factors in pediatric surgery, as they take place in a very sensitive part of the postoperative phase. Positive changes in those processes could be seen as motivating factors by those involved. Furthermore, both patients and their parents encountered challenges when attempting to make an unbiased evaluation of the quality of care. As previously mentioned, they often presumed that healthcare professionals possessed a certain level of expertise without scrutinizing it objectively. The assessment was therefore often based on whether the procedure went smoothly, staff was friendly, interventions were explained and fears were recognized and taken away.

Improvements to the dressing change procedures may also increase the overall satisfaction of the medical personnel. The international nursing study RN4Cast performed in 12 European countries, with 33,659 interviewed nurses in 488 hospitals, demonstrated the dissatisfaction of the skilled nurses. The European average was dissatisfaction with 25%, but German nurses, with 37%, were considerably more dissatisfied than the average, and 36% of German nurses were willing to change employers within the next year. Overall, 53% of those willing to change considered switching to another professional branch [15]. In hospitals and other medical areas, many skilled nurses are subjected to difficult working conditions, such as shift work, incompatibility of work and profession, insufficient remuneration and a lack of further training opportunities [16]. In addition, 32% of the German population will be over 65 years old in 2040. At the same time, there are fewer and fewer young people in the field of health and nursing, and from 1995 to 2008, the number fell by 16.6% [17]. Thus, hospitals tried to improve staff satisfaction and working conditions. The increase in satisfaction of the medical professionals involved in dressing changes is a step towards making everyday work more pleasant through improved processes in a frequent daily routine, such as dressing changes, especially in times of difficult working conditions and overall general job dissatisfaction among skilled nursing staff.

### 4.4. Dressing Types and Sizes

The size of the dressing changes was divided into three categories: small plaster dressing, large plaster dressing and other dressing material based on the size of the wound area. The latter combined all specific dressings, such as VAC dressings. A more distinct classification might help to obtain differentiated results within the individual groups. However, since the majority of the dressing changes (n = 83) could be divided very well into small and large and only a small proportion (n = 17) was recorded as special logistically or time-consuming dressing changes, the results demonstrated the effects of the dressing trolleys in the respective groups.

The number of plasters/bandages that was allocated to which group was defined beforehand. The exact amount of dressing material was therefore not recorded. A more precise classification of the dressing changes could have provided even more detailed results, which can be performed in following studies with larger groups. This might lead to the identification of procedures and dressing types from which children and caregivers might benefit the most.

### 4.5. Dressing Change Preparation Time

Particularly during the data collection of group 1, the preparation time could not be recorded in n = 20 cases. The preparation time during this period included various tasks, such as gathering the materials from the warehouse or the treatment room and the subsequent preparation.

In order to obtain even more detailed results, a differentiated survey of these individual times and storage locations would be valuable.

The setting of an observational study made this type of implementation more difficult, as the specialist nurses had already carried out the preparation in part in their daily routine before study group was informed about the upcoming dressing changes by the medical colleagues.

### 4.6. Duration of the Dressing Change

The time between removing the first patch and sticking the last patch onto the patient was defined as the duration of the dressing change. A division of the procedure into different phases (loosening the plaster, inspection, cleaning the wound, wound care and application of the new dressing) could have led to more precise data as to which part of the procedure had undergone the greatest change due to the use of the dressing trolley. But the entire specialist nursing medical staff will then have to be made aware that the respective steps have to be announced at each dressing change so that they could be recorded. In addition, the time intervals to be recorded would have been very short, as many of the procedures carried out were performed very fast.

### 4.7. Subjective Evaluation of the Dressing Trolley by a Questionnaire

For creation of the questionnaire, the following aspect had to be considered: The questions asked should support or refute the hypothesis associated with the questionnaire, which was “Application of dressing trolleys increases the satisfaction of the people involved and the logistical and temporal components of the dressing change”.

The type of questions chosen is important. closed and open questions both have different advantages and disadvantages. Respondents are able to fill out open questions completely “freely”. The answer therefore reflects the actual opinion of the person questioned and is not a compromise based on the answer options presented to them. However, answers to open questions are heavily dependent on the skills of the answering individual. Many respondents may find it difficult to answer open questions and a great effort is involved in collecting and evaluating those data. Closed questions are the gold standard for data collection. They can be answered quickly and specifically and can also be quickly recorded and evaluated [18,19]. Their disadvantage is that respondents occasionally do not find themselves in the given answer categories. It could therefore happen that respondents do not answer the questions, make deliberate false statements or tick something that was only partially applicable: We decided for a questionnaire with closed single answer questions because of the advantages mentioned above.

The German school grading system of 1–6 was chosen as the answer categories. These are ordinally scaled values that have a natural ranking. By using the school grading system, which is known and widespread in Germany, it ensured that the assessment could be carried out quickly.

The questionnaires were answered by the staff who were responsible for handling the dressing trolley during the dressing change. In the case of very anxious children, it was necessary for the specialist nurse to look after the patient very intensively and calming them down. In this case, medical colleagues or students took over the handling of the dressing trolley. It was noted on the questionnaire so that any differences between the two professional groups could be analyzed.

In addition to the requirements for the creation of questions in a scientific questionnaire, the quality criteria of a scientific test should also apply to the entire questionnaire and the survey.

### 4.8. Limitations

As this study was conducted in an prospective observational manner, the following limitations apply: selection bias due to a lack of randomization, overestimation of the treatment effect due to a lack of randomization, influences from known and unknown disruptive factors (confounders), no possibility of blinding, incomplete recording of all relevant data, and low internal and high external validity. In contrast, a study by Benson and Hartz showed that there was no significant difference between results when investigating the same question in a randomized controlled and an observational study [20].

The trolleys were introduced into the division of pediatric surgery. The patient clientele was limited to children with an median age of 2.5 years and bandage changes after classic pediatric surgery. A wider range of patients and surgical disciplines would have been desirable in order to be able to provide more informative values. 

Concerning limitations of the questionnaire, suitable answer categories might not have been available for all respondents. We tried to eliminate for confounders by using the median of all answers. Drawing conclusions about the evaluation of dressing trolleys in other departments seems difficult due to the very different loading of the trolleys and the spatial and logistical differences. However, this questionnaire reveals how dressing trolleys were generally assessed by the staff. 

One relevant limitation may be that patient and caregivers satisfaction was determined as an outcome parameter in our study. Satisfaction is a subjective sensation. Despite objectification by a visual analogue scale, it could not be completely ruled out that other factors in connection with the dressing changes may have influenced patient and caregiver satisfaction. For this purpose, an additional, detailed questionnaire with different aspects of dressing change satisfaction would have been valuable. In the context of a clinical observational study, the integration of a detailed questionnaire could not be implemented without far-reaching problems. On the one hand, there would have been an additional time burden for the staff involved, which in turn could have reduced the acceptance of the study, as acceptance and cooperation of the medical and nursing staff was essential for the collection of objective results in this study.

A significant limitation associated with the primary use of dressing trolleys pertains to hygiene considerations and infection rates. In the United States, these trolleys were discontinued in the 1940s due to increased infection rates observed when they were commonly utilized in surgical wards [3,21,22]. One may argue that the presence of contamination on the trolley and its components could still potentially contribute to higher infection rates in modern healthcare settings. However, contemporary practices involve the preparation of trolleys with dressings and instruments individually packaged by a central supply [22]. This method ensures that the act of opening a package, whether it is stored in a warehouse or within a dressing trolley, adheres to the same hygienic principles. Of course, proper hand hygiene and surface disinfection of the trolley are prerequisites, as emphasized by Taylor et al. in 1962 [23]. It is plausible to hypothesize that the increased focus on proper hand hygiene during the ongoing COVID-19 pandemic may lead to reduced infection rates associated with dressing changes. However, it is important to note that our study was not specifically designed to test this hypothesis. As this study was conducted to evaluate the satisfaction of the patients, caregivers and parents, as well as logistical aspects, we did not focus on postprocedural complications, such as infection rates in our secondary outcome parameters. Nevertheless, though not depicted in this manuscript, we re-evaluated the procedures and did not find a relevant rate of wound infections in both groups.

Performing this study with patients of different age groups and after interventions from different surgical specialties would have been desirable. In addition, more dressing changes could have been investigated. These two points would have provided increased informative value with regard to dressing changes in all surgical disciplines. Further studies have to take this into account.

## 5. Conclusions

Our findings unequivocally demonstrate a notable augmentation in caregiver and parental satisfaction, albeit not among the children themselves, following the implementation of a dedicated dressing trolley for conducting bedside dressing changes within the patients’ rooms. Moreover, the utilization of this dressing trolley exhibited improvements in various logistical and time-dependent aspects of the dressing procedures.

It is crucial to emphasize that any direct comparison with previously reported studies is precluded, as our investigation represents the pioneering study to comprehensively investigate the effects of a dressing vehicle specifically in the realm of pediatric surgery.

The comprehensive enhancement in logistical operations, coupled with the subjective satisfaction reported by both caregivers and personnel involved in dressing changes, strongly advocates for the adoption of mobile dressing trolleys in pediatric departments that have yet to embrace this practice. Not only does the introduction of such trolleys heighten the satisfaction levels of nearly all parties involved, but it also enhances the efficiency and expediency of the procedures from an economic standpoint. Ultimately, this advancement in dressing practices contributes to the overall reduction in the invasiveness of surgical interventions, aligning with the ultimate goal of facilitating minimally invasive surgery and minimally invasive postoperative care.

## Figures and Tables

**Figure 2 children-10-01089-f002:**
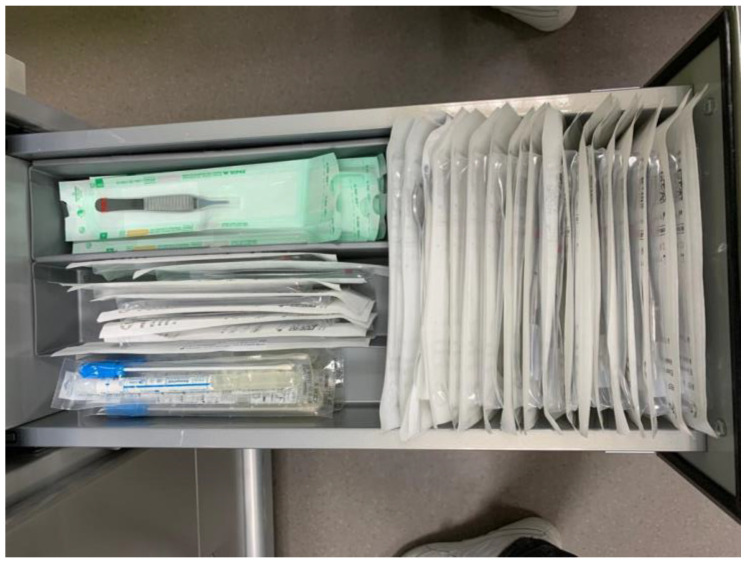
An opened drawer (no. 2) of the trolley displaying the sterile and sealed instruments used for the dressing changes.

**Figure 3 children-10-01089-f003:**
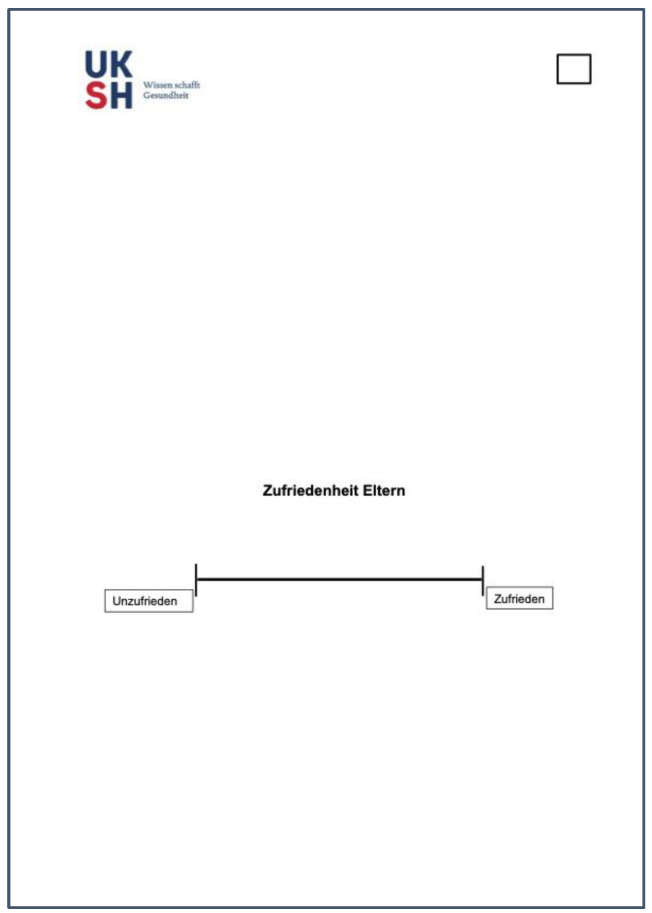
The visual analogue scale used.

**Table 1 children-10-01089-t001:** Contents of the dressing trolley.

Drawer No.	Contents of the Drawer
1	Thread removal materials, syringes of different sizes, sutures and utensils for wound irrigation
2	Sterile scissors, tweezers and clamps (Figure 2)
3	Sterile gloves and sterile drapes in different sizes
4	Rewards for brave children and Moltex underlays
5	Suction compresses, small Mepithel (Mölnlycke, Germany), elastic bandages, cotton bandages, disposable gowns and face masks
6	Large Mepithel (Ethicon, Germany), elastic bandages, cotton wool bandages disposable gowns, mouthguards, gauze compresses, gauze bandages, Steristrips, Cavilon skin protection, and antiseptic skin disinfectants
7	Wound dressings of different sizes, taping strips and plaster remover

**Table 2 children-10-01089-t002:** Questionnaire concerning dressing trolleys characteristics with the instructions: “For the respective questions, please tick one of the numbers between 1 (very good)–6 (insufficient).”

Question	1	2	3	4	5	6
How did you find the selection of bandage materials?						
How did you find the number of bandages?						
How did you find the manageability of the dressing trolley?						
How did you find the cleanliness of the dressing trolley?						
Did the dressing trolley facilitate your work?						
What is the probability that you will use the dressing trolley in the future?						

**Table 3 children-10-01089-t003:** Results—baseline, primary and secondary outcome parameters. ^$^: Median (interquartile range of Q25/Q75), *: n (%), ^$$^: Mann–Whitney-U test, ^&&^: contingency tables and chi-square test.

	Group 1 No Trolley	Group 2 with Trolley	*p*-Value
**n**	51	49	
**Age (years) ^$^**	1.97 (0.21/9.65)	2.64 (0.57/10.05)	0.41 ^$$^
**Sex**			0.68 ^&&^
Female *	26 (51)	27 (55.1)	
Male *	25 (49)	22 (44.9)	
**The child’s mood (VAS, mm) ^$^**	81.5 (60.0/96.5)	86.0 (70.75/95.125)	0.57 ^$$^
Missing data	4	3	
**Gender of the accompanying parent**			0.077 ^&&^
Female *	28 (54.9)	36 (73.5)	
Male *	13 (25.5)	10 (20.4)	
No accompanying parent present *	10 (19.6)	3 (6.1)	
**Category of the surgical procedure**			0.2 ^&&^
1. Single incision laparoscopic surgery (SILS) *	2 (3.9)	0 (0.0)	
2. Multiport laparoscopy/robotics *	1 (2.0)	6 (12.2)	
3. Umbilical minilaparotomy *	5 (9.8)	2 (4.1)	
4. Median/transverse laparotomy *	22 (43.1)	19 (38.8)	
5. Inguinal hernia, lymph node biopsy/abscess drainage, (sub)cutaneous tumor extirpation/biopsy, burns, anorectal malformation repair *	14 (27.5)	16 (32.7)	
6. Vacuum-assisted closure (VAC) *	7 (13.7)	6 (12.2)	
**Type of dressing applied according to the size of the wounds**			0.98 ^&&^
Small dressings with a size of up to 5 cm and a maximum of 3 dressings used *	25 (49.0)	25 (51.0)	
Large dressings of at least 12 cm (e.g., median laparotomy) *	17 (33.3)	16 (32.7)	
Special types of dressings comprise those that could not be included in the two abovementioned categories (e.g., VAC) *	9 (17.6)	8 (16.3)	
**Room in whichthe dressing change was performed**			0.85 ^&&^
The patients room *	43 (84.3)	42 (85.7)	
The examination room *	8 (15.7)	7 (14.3)	
**Sedation of the child for the dressing change**			0.58 ^&&^
No sedation *	49 (96.1)	48 (98.0)	
Sedation *	2 (3.9)	1 (2.0)	
**Multiresistant Bacteria**			0.16 ^&&^
Patients without multiresistant bacteriae *	50 (98.0)	45 (91.8)	
Patients with multiresistant bacteriae *	1 (2.0)	4 (8.2)	
**Preparation time (minutes)**	1:16 (0:32/3:43)	0:00 (0:00/0:10)	**<0.001 ^$$^**
**Duration of the dressing change (minutes)**	3:42 (1:48/8:09)	1:46 (1:05/7:53)	0.05 ^$$^
**Frequency of leaving the patient’s room to retrieve missing materials**			**0.022 ^&&^**
Never left the patients room to retreive missing materials *	26 (51)	41 (83.7)	
Left once	17 (33.3)	5 (10.2)	
Left 2 times *	5 (9.8)	2 (4.1)	
Left 3 times *	1 (2.0)	1 (2.0)	
Left 4 times *	1 (2.0)	0 (0.0)	
Left 5 times *	0 (0.0)	0 (0.0)	
Left 6 times *	1 (2.0)	0 (0.0)	
**Time spent outside the patient’s room for retrieving missing materials (seconds)**	46 (27–117)	18.5 (13.5–85)	**0.048 ^$$^**
**Total duration of the dressing change consisting of its preparation, execution and cleanup (minutes) ^$^**	7:04 (3:45–20:26)	1:55 (1:05–7:58)	**0.001 ^$$^**
Missing data	21	0	
**Satisfaction of the surgeons with the dressing change (VAS, mm) ^$^**	85.0 (65.0–96.5)	97 (88.5–98.5)	**<0.001 ^$$^**
**Satisfaction of the parents with the dressing change (VAS, mm) ^$^**	94 (75–98)	96.5 (94–98.125)	**0.042 ^$$^**
Missing data	12	7	
**Satisfaction of the nursing staff with the dressing change (VAS, mm) ^$^**	81 (58.5–90.0)	90.25 (IQR 77.375–97.875)	**0.015 ^$$^**
Missing data	8	13	
**Satisfaction of the observer with the dressing change (VAS, mm) ^$^**	79.5 (69.5–87)	93 (90.75–96)	**<0.001 ^$$^**
**Satisfaction of the children, ifolder than 12 years, with the dressing change (VAS, mm) ^$^**	87.5 (77.0–94.25)	93.25 (70.875–96.75)	0.7 ^$$^
Missing data	42	41	

**Table 4 children-10-01089-t004:** Overall assessment of the dressing trolley.

	N	Minimum	Maximum	Mean
Selection of bandage material?	47	1	3	1.34
Number of bandages?	47	1	2	1.09
Manageability?	47	1	3	1.60
Cleanliness?	47	1	5	1.21
Work facilitation?	47	1	6	1.70
Probability that you will use the dressing trolley in the future?	47	1	5	1.45
Overall grade				1.40

## Data Availability

The obtained data is unavailable due to privacy or ethical restrictions and thus not available to the public.

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
