# Peer review of "Mobile Dressing Trolleys Improve Satisfaction and Logistics on Pediatric Surgery Wards"

_children, 2023, doi:10.3390/children10071089_

Round 1

Reviewer 1 Report (New Reviewer)

Dear Authors, 

Please provide photos of the trolley and the dressing used for this study. This will help readers to understand the manuscript better.

kind regards,

Reviewer

Author Response

Reviewer 1:

Dear Authors, Please provide photos of the trolley and the dressing used for this study. This will help readers to understand the manuscript better.

We greatly appreciate your recommendation. We have incorporated the following photos:

Phtotograph of the trolley used in this study. The trolley is fabricated with metal which improves the disinfection, also note the desinfection calender on the short side of the trolley. The Contents are stored in drawers and sealed, see Figure 2.

An opened drawer of the trolley displaying the sterile and sealed instruments used for the dressing changes.

Reviewer 2 Report (New Reviewer)

The authors made a prospective observational non randomized study named: “Mobile dressing trolleys improve satisfaction and logistics on 2 pediatric surgery wards “

The paper is well written, both linguistically and well as by contents.

All except concern on the end are minor remarks:

1.       METHODS: Please clearly state inclusion criteria like exclusion criteria are stated! Additionally, state primary and secondary outcomes of the study.

2.       METHODS: my question goes: did the nonrandomized selection was conducted by staff preference or was it done in a manner: first group 1 (cca n=50) then next 50 were into group 2. Or there was some in group 1 and some in group 2, at the same time?

3.       METHODS: Sentence is as follows: “The general satisfaction with the dressing change (visual analogue scale, VAS, 0-100mm) of the parents, children (if older than 12 years), nursing staff, surgeons and the observer.” Can authors put a detailed explanation what is this type of VAS scale 0-100 mm? Also, please add a visual (picture) example of an VAS. Where there any children older then age of 12 years?

4.       RESULTS: median age was 1.97 years (interquartile rage IQR: 0.21 - 9.65) and 2.64 (IQR: 0.57 - 10.05) years in groups 1 and 2 respectively. The age group seems too immature, for the objectivize the VAS scale - mood. Nevertheless, this may be appropriate and correct, but then I would ask the authors to find some reference on this young children and usage of that VAS scale.  

5.       Where there any complications? infection rates, SSI?

6.       I would advise putting the satisfaction with the procedures, dressing change, duration, etc. into proper table, with proper p values, for it will be more reader friendly.

7.       My biggest concern here is the relatively subjective nature of the study, because the wound dressing much differs based on type of surgery [especially for the duration (seconds) of the dressing change] the child underwent. It would be much better if the same pathologies/surgeries were compared in all observed and collected data. The data would be false and wrongly interpreted if in one group there are more laparotomies operations, and in other one laparoscopic ones, it is hard to compare the dressing of wounds in different types of surgeries… Otherwise I feel (and this is also a benign remark as well as a subjective one) the total number of 100 (cca 50 by group) patients (children in diverse age groups) is too small to derive such conclusions and should be enlarged. For all this I would strongly advise for the statistician to give his/her professional opinion on sample size and evidence power. 

Author Response

Reviewer 2:

The paper is well written, both linguistically and well as by contents.

Thank you very much for your positive comment.

All except concern on the end are minor remarks:

1. METHODS: Please clearly state inclusion criteria like exclusion criteria are stated! Additionally, state primary and secondary outcomes of the study.

Thank you very much for your comment. We added the following paragraph to the methods section, pertaining the inclusion criteria: “This study involved the inclusion of pediatric surgery inpatients, specifically children below 18 years of age, who required postoperative wound dressing changes. Exclusion criteria encompassed cases where patient or parental consent was lacking, children staying in the intensive care unit, dressing changes performed in the operating room under general anesthesia, and dressings, such as skin glues, that did not necessitate regular dressing changes.

The primary and secondary outcomes of this study were added in the manuscript in the Data Section: “

Outcome Parameters and Data acquisition

The primary outcome parameter was the subjective satisfaction with the dressing change of the accompanying parent (visual analogue scale, VAS, 0-100mm) of the parents with or without application of the dressing trolley. Secondary outcome parameters were collected as the following data:

The gender of the patient, the gender of the parents present, the type and date of the surgery.

The anticipated mood of the child undergoing the dressing change, taken after the procedure by a visual analog scale of the parents and, if the parents were not present, the specialist nurses.

The general satisfaction with the dressing change (visual analogue scale, VAS, 0-100mm) of the children (if older than 12 years), nursing staff, surgeons and the observer. The left and right end the VAS was labeled with the words "unsatisfied" and "satisfied", respectively.

The time (seconds) for the preparation of the dressing change: Only the time spent looking for and depositing the dressing materials was recorded. The time taken to cover the distance between the store rooms and the patient's room was not recorded because the dressing trolley was also stationed in the store rooms and had to cover the same distance when moved into the patients rooms.

The duration (seconds) of the dressing change

The occurrences (n) the personnel had to leave the patients room to fetch additional material currently missing for the dressing change.

The duration (seconds) the personnel was outside the room to catch up on missing material.

The occurrences (n) of any interruption of the dressing change.

During time period two, a questionnaire concerning the general load, cleanliness, manageability and overall evaluation of the dressing trolley was answered by the medical staff performing the dressing change. The questions were rated according to the German school grading system 1 (very good) to 6 (bad). (Table 2)

The use of the dressing trolley (yes / no), the location of the dressing change (patient's room, examinaton room) and the surgeon performing the dressing change.

The type of the applied dressing was classified according to the size of the wounds: Small dressings with a size of up to 5 cm and a maximum of 3 of those (e.g. for laparoscopic procedures). Large dressings of 12 cm (e.g. median laparotomy). Special types of dressings contained those that could not be included in one of the first two categories (e.g. VAC).

The category of the surgical procedure (1 = Single incision laparoscopic surgery (SILS); 2 = Multiport laparoscopy / robotics; 3 = Umbilical minilaparotomy; 4 = Median / transverse laparotomy; 5 = Inguinal hernia, lymph node biopsy / abscess drainage, sub/cutaneous tumor extirpation / biopsy, burns, anorectal malformation repair; 6 = Vacuum Assisted Closure (VAC) )

2. METHODS: my question goes: did the nonrandomized selection was conducted by staff preference or was it done in a manner: first group 1 (cca n=50) then next 50 were into group 2. Or there was some in group 1 and some in group 2, at the same time?

We sincerely appreciate your significant comment. Indeed, our selection process followed a non-randomized approach based on the chronological order of patient surgeries in our department. Initially, we allocated the first 50 patients to group one, where the trolley was not utilized. Subsequently, upon introducing the trolley into our clinical practice, we included the remaining half of our cohort in the group that underwent dressing changes using the trolley. Thus, the two groups were acquired during distinct time periods.

We have added the following to our methods section “Observational Periods”: “Therefore, the two groups were acquired during distinct time periods.

3. METHODS: Sentence is as follows: “The general satisfaction with the dressing change (visual analogue scale, VAS, 0-100mm) of the parents, children (if older than 12 years), nursing staff, surgeons and the observer.” Can authors put a detailed explanation what is this type of VAS scale 0-100 mm? Also, please add a visual (picture) example of an VAS. Where there any children older then age of 12 years?

We sincerely appreciate your importand comment. We solely employed the visual analogue scale (VAS) in patients to assess satisfaction in children aged 12 years and above. Consequently, we lacked any data regarding VAS-measured satisfaction in children below 12 years of age. Among the participants, there were 17 children who met the age criterion of 12 years or older. In group 1, comprising 51 individuals, the VAS-measured satisfaction for dressing change was 87.5 (interquartile range: 77.0 - 94.25), with 9 participants reporting satisfaction. In group 2, consisting of 49 individuals, the VAS-measured satisfaction was 93.25 (interquartile range: 70.875 - 96.75), with 8 participants reporting satisfaction. Further details can be found in the results section.

Please see the attached picture of the VAS that we used:

4. RESULTS: median age was 1.97 years (interquartile rage IQR: 0.21 - 9.65) and 2.64 (IQR: 0.57 - 10.05) years in groups 1 and 2 respectively. The age group seems too immature, for the objectivize the VAS scale - mood. Nevertheless, this may be appropriate and correct, but then I would ask the authors to find some reference on this young children and usage of that VAS scale. 

We sincerely appreciate your highly valuable comment. As previously stated, the satisfaction (VAS Score) of children below 12 years of age was not assessed in our study. However, we did evaluate the mood of the child following the dressing change procedure. The anticipated mood of the child was assessed using a visual analog scale (VAS), which was completed by the parents. In cases where the parents were absent, the assessment was conducted by specialist nurses.

5. Where there any complications? infection rates, SSI?

We sincerely appreciate your highly valuable comment, particularly regarding the utilization of dressing trolleys in various domains of infectious complications, including wound and surgical site infections. This study aimed to assess the satisfaction levels of patients, caregivers, and parents, along with logistical aspects. Consequently, our secondary outcome parameters did not primarily focus on post-procedural complications, such as infection rates. Nonetheless, it is important to note that, although not explicitly presented in this manuscript, we thoroughly reevaluated the procedures and found no significant occurrence of wound infections in either of the study groups.

We have added the following into the discussion section of the limitations of our study: “As this study was conducted to evaluate the satisfaction of the patients, caregivers and parents, as well as logistical aspects, we did not focus on post procedural complications, such as infection rates in our secondary outcome parameters. Nevertheless, not depicted in this manuscript, we reevaluated the procedures and did not find a significant rate of wound infections in both groups.”

6. I would advise putting the satisfaction with the procedures, dressing change, duration, etc. into proper table, with proper p values, for it will be more reader friendly.

Thank you very much for this comment, we changed the display of the data in a table.

7. My biggest concern here is the relatively subjective nature of the study, because the wound dressing much differs based on type of surgery [especially for the duration (seconds) of the dressing change] the child underwent. It would be much better if the same pathologies/surgeries were compared in all observed and collected data. The data would be false and wrongly interpreted if in one group there are more laparotomies operations, and in other one laparoscopic ones, it is hard to compare the dressing of wounds in different types of surgeries. Otherwise I feel (and this is also a benign remark as well as a subjective one) the total number of 100 (cca 50 by group) patients (children in diverse age groups) is too small to derive such conclusions and should be enlarged. For all this I would strongly advise for the statistician to give his/her professional opinion on sample size and evidence power.

Thank you for your valuable comment. You have raised a crucial point, and we would like to refer to the table we have constructed based on your sixth comment. In our study, we categorized the types of surgeries undergone by the children into five distinct categories: 1 = Single incision laparoscopic surgery (SILS); 2 = Multiport laparoscopy / robotics; 3 = Umbilical minilaparotomy; 4 = Median / transverse laparotomy; 5 = Inguinal hernia, lymph node biopsy / abscess drainage, sub/cutaneous tumor extirpation / biopsy, burns, anorectal malformation repair; 6 = Vacuum Assisted Closure (VAC).

We found no significant difference in the frequency and distribution of surgery types between group 1 and group 2. Additionally, we documented the types of dressings used, as shown in the table, too. There was also no discrepancy in the sizes of dressings used, whether small, large, or special ones, between groups 1 and 2.

Based on these findings, we are confident that neither the type of surgery nor the size of the dressing has a significant impact on the VAS scales or the duration of dressing changes. However, we acknowledge your suggestion that a larger sample size and stratification based on different types of surgery could provide more valuable insights, especially in detecting subtle differences.

Regarding the calculation of the sample size, we conducted a pre-study analysis, that indicated that each group required a sample size ranging from 26 to 64 children, taking into consideration the desired statistical power, effect size, and alpha-level, which had to be assumed as no prior data can be found on this subject.

We hope to have answered all your questions.

Round 2

Reviewer 2 Report (New Reviewer)

Authors answered all questions and made the proper changes to what was noted.

This manuscript is a resubmission of an earlier submission. The following is a list of the peer review reports and author responses from that submission.

Round 1

Reviewer 1 Report

Dear Franck and colleagues

Congratulations on the manuscript you submitted. I appreciate the effort you put into your work and the opportunity to review it. After reading your manuscript, I must say that I have some remarks.

Firstly, the title is too large. I suggest shortening it. 

The introduction section lacked a clear and concise summary of the existing literature, which made it hard to understand the motivation behind the study.

The phrases in Table 1 must be in sentence case.

The authors pointed out that the alpha level was 0.05 for the calculation of sample size, but it is not clear what p-value cutoff was considered significant for the main analysis. In line 207, the authors state that “The result was statistically just significant (p = 0.05)”. If the authors considered p-value < 0.05 was considered significant (like most of the research does), then 0.05 would be considered not significant. 

The conclusion section is too extensive and not straight, with vague statements. The conclusion should answer the study’s objectives written in the introduction section. 

Thank you again for the opportunity to review your manuscript.

Author Response

Reply to the Reviewers

Dear Madam and Sir,

 thank you very much for the opportunity to revise our manuscript. Please find attached our answers and corrections to the reviewers comments.

Editor:

Unfortunately this has been tried and was standard of care in the US and has been shown to increase infection rates. The authors did not discuss infection rates. Yes, it is more efficient, the authors have shown that nicely, however, they did not address the risk of infection that has banished similar systems from current use. I would encourage the authors to stratify clean/clean contaminated/contaminated/dirty wounds with and without the trolley system as hand hygiene has likely improved drastically since the beginning of the pandemic and transmission rates could have fallen using this system.

Thank you for your important comment and suggestion. We fully concur with the significance of hygiene as a crucial determinant in the utilization of dressing trolleys. The discussion you referenced regarding hygiene considerations dates back to 1959.1 Notably, there have been substantial advancements in hygiene practices since then. In our study, the dressing trolley employed adhered to the guidelines set forth by Myers in 1959, which emphasized the use of individually packaged supplies to ensure the administration of uncontaminated dressings and instruments.1,2

While our primary focus was to evaluate the impact of the trolley on staff and patient satisfaction, as well as logistical aspects, infectious parameters were not directly assessed as part of this study. Instead, such parameters were collected solely as part of our routine clinical morbidity and mortality evaluation. However, we categorized the surgical procedures into different groups, specifically groups 5 and 6, which encompassed cases involving potentially infected, contaminated, or "dirty" wounds. This allowed for a form of stratification as you recommended. Nevertheless, analyzing each group individually did not yield any different outcomes. Accordingly, we have incorporated the following paragraphs into the limitations section of our manuscript based on your input:

"A significant limitation associated with the primary use of dressing trolleys pertains to hygiene considerations and infection rates. In the United States, these trolleys were discontinued in the 1940s due to increased infection rates observed when they were commonly utilized in surgical wards.1–4 One may argue that the presence of contamination on the trolley and its components could still potentially contribute to higher infection rates in modern healthcare settings. However, contemporary practices involve the preparation of trolleys with dressings and instruments individually packaged by a central supply.1 This method ensures that the act of opening a package, whether it is stored in a warehouse or within a dressing trolley, adheres to the same hygienic principles. Of course, proper hand hygiene and surface disinfection of the trolley are prerequisites, as emphasized by Taylor et al. in 1962.5 It is plausible to hypothesize that the increased focus on proper hand hygiene during the ongoing COVID-19 pandemic may lead to reduced infection rates associated with dressing changes. However, it is important to note that our study was not specifically designed to test this hypothesis."

Thank you once again for your valuable input, which has greatly contributed to the improvement of our manuscript.”

Reviewer 1:

Firstly, the title is too large. I suggest shortening it.

Thank you very much for your recommendation. We changed the title to: “Mobile dressing trolleys improve satisfaction and logistics on pediatric surgery wards

The introduction section lacked a clear and concise summary of the existing literature, which made it hard to understand the motivation behind the study.

Thank you very much for your supportive comments. We changed to introduction to: “Undergoing surgical intervention is an arduous event that impacts both children and their caretakers. The subsequent phase of postoperative recovery, which involves the management of wounds and the changing of dressings in delicate and potentially painful areas, plays a pivotal role in the overall healing process.[1] While efforts have been made to mitigate perioperative stress in children, the specific impact of dressing changes on pediatric patients remains unexplored.[2]

Literature has touched upon the utilization of dressing trolleys or carts for wound care, with discussions on hygiene aspects dating back to as early as 1958.[3,4] However, pertinent data pertaining to the utilization of dressing trolleys in the context of pediatric surgery is currently absent. Consequently, the objective of this study was to assess the subjective satisfaction levels of pediatric patients and their caregivers both prior to and following the implementation of a specially designed pediatric surgical dressing trolley. Additionally, the study aimed to determine whether the introduction of such a trolley would enhance the logistical aspects of these procedures.”

The phrases in Table 1 must be in sentence case.

Thank you very much for your help, we changed Table 1 accordingly.

The authors pointed out that the alpha level was 0.05 for the calculation of sample size, but it is not clear what p-value cutoff was considered significant for the main analysis. In line 207, the authors state that “The result was statistically just significant (p = 0.05)”. If the authors considered p-value < 0.05 was considered significant (like most of the research does), then 0.05 would be considered not significant.

Thank you very much for this very important comment. The p-value considered as significant was p<0.05 – we have added this to the Materials and Methods Section. Just as you said, we also changed line 207 into the right result: “The result was statistically just NOT significant (p = 0.05)”

The conclusion section is too extensive and not straight, with vague statements. The conclusion should answer the study’s objectives written in the introduction section.

Thank you very much for your comment – we changed the conclusion section to:

Our findings unequivocally demonstrate a notable augmentation in caregiver and parental satisfaction, albeit not among the children themselves, following the implementation of a dedicated dressing trolley for conducting bedside dressing changes within the patients' rooms. Moreover, the utilization of this dressing trolley exhibited improvements in various logistical and time-dependent aspects of the dressing procedures.

It is crucial to emphasize that any direct comparison with previously reported studies is precluded, as our investigation represents the pioneering study to comprehensively investigate the effects of a dressing vehicle specifically in the realm of pediatric surgery.

The comprehensive enhancement in logistical operations, coupled with the subjective satisfaction reported by both caregivers and personnel involved in dressing changes, strongly advocates for the adoption of mobile dressing trolleys in pediatric departments that have yet to embrace this practice. Not only does the introduction of such trolleys heighten the satisfaction levels of nearly all parties involved, but it also enhances the efficiency and expediency of the procedures from an economic standpoint. Ultimately, this advancement in dressing practices contributes to the overall reduction in the invasiveness of surgical interventions, aligning with the ultimate goal of facilitating minimally invasive surgery and minimally invasive postoperative care.

We hope to have answered all your questions.

We would be very happy if you accept our revised manuscript for publication in your journal.

Prof. Dr. med. Robert Bergholz 

UKSH Campus Kiel

References

  1. Myers, R. S. How to keep dressing cart from carrying contamination. Mod. Hosp. 93, 136 (1959).
  2. Graves, C. L., Longley, E. K., Erickson, E. H. & Debusk, R. W. Where, oh where has our dressing cart gone? Mod. Hosp. 80, 60–61 (1953).
  3. Dingman, R. O., Natvig, P. & Winkler, J. M. A new dressing cart for plastic surgery. Plast. Reconstr. Surg. 1946 19, 72–77 (1957).
  4. [The dressing cart]. Soins Rev. Ref. Infirm. III–IV (1989).
  5. Taylor, G. W., Shooter, R. A. & Allen, S. M. Ward dressing technique for use with central supply. Br. Med. J. 2, 1746–1747 (1962).
